# Cas9+ conditionally immortalized neutrophil progenitors as a tool for genome-wide CRISPR screening for neutrophil differentiation and function

**Robyn M Jong[1,2], Krystal L Ching[1], Nicholas E Garelis[1], Alex Zilinskas[1], Xammy Huu Wrynla[1,2], Sagar Rawal[1], Bianca C Hill[1], Bridget A Luckie[1], Lillian Shallow[1], Jeffery S Cox[1], Gregory M Barton[1], Sarah A Stanley[1,2]***

[1]Department of Molecular and Cell Biology, Division of Immunology and Pathogenesis, University of California, Berkeley, Berkeley, United States; [2]School of Public Health, Division of Infectious Diseases and Vaccinology, University of California, Berkeley, Berkeley, United States

**Abstract** Neutrophils are short-lived cells of the innate immune system that play numerous roles in defense against infection, regulation of immune responses, tissue damage and repair, autoimmunity, and other non-communicable diseases. Understanding neutrophil function at a mechanistic level has been hampered by the difficulty of working with primary neutrophils, which die rapidly upon isolation, and the relative paucity of neutrophil cell lines. Here, we report the creation of a Cas9 +ER-Hoxb8 neutrophil progenitor cell line that enables both forward and reverse genetic analysis of neutrophils. By editing progenitors via transduction with sgRNAs and then withdrawing estrogen, Cas9-edited neutrophils are produced with high efficiency. Importantly, neutrophil differentiation of edited progenitors occurs both in vitro in cell culture and when transferred into murine recipients. To demonstrate the utility of Cas9 +ER-Hoxb8 progenitors for forward genetics, we performed a pooled CRISPR screen to identify factors required for survival during neutrophil differentiation. This screen identified hundreds of genes, including *Cebpe*, a transcription factor known to be required for neutrophil differentiation from pre-neutrophils to immature neutrophils. Using this progenitor cell line, we confirmed that *Cebpe* is required for neutrophil differentiation in vivo, validating the utility of this line for studying in vivo phenotypes. The screen also identified all components of the WASH complex as being required for neutrophil differentiation, extending its known role in hematopoietic stem cell differentiation to later stages of neutrophil development. Taken together, this resource enables the analysis of the role of neutrophils in numerous disease states using genetics for the first time.

**\*For correspondence:**
sastanley@berkeley.edu

**Competing interest:** The authors declare that no competing interests exist.

## Editor's evaluation

Jong et al. provide and convincingly validate a resource for performing CRISPR screenings to study neutrophil differentiation and function by generating Hoxb8 cells that constitutively express Cas9. This valuable library-screening approach has the potential to improve on the established lentiviral CRISPR-Cas9 editing of Hoxb8 cells.

## Introduction

Neutrophils, also known as polymorphonuclear cells, are the most abundant circulating immune cells in humans. Neutrophils are short-lived cells that are produced in the bone marrow and are rapidly recruited to sites of infection, where they eliminate pathogens utilizing a potent antimicrobial

arsenal that includes production of reactive oxygen species (ROS), release of granules containing antimicrobial proteins into the extracellular or phagosomal space, and production of neutrophil extracellular traps (NETs) (*Amulic et al., 2012*). Humans with primary neutrophil deficiencies are susceptible to infection with *Aspergillus fumigatus, Candida albicans, Staphylococcus aureus, Burkholderia cepacia, Nocardia,* and other bacterial and fungal species (*Dinauer, 2016*). However, the role of neutrophils in human biology is not as simple as only participating in defense against infection. For example, neutrophils are not effective against every pathogen. In the case of *Mycobacterium tuberculosis,* neutrophils are not able to kill the bacteria and their recruitment to sites of infection exacerbates inflammation to the detriment of the host (*Ji et al., 2020*; *Mishra et al., 2017*; *Nair et al., 2018*; *Nandi and Behar, 2011*; *Ravesloot-Chávez et al., 2021*). Indeed, neutrophils often play a role in tissue damage associated with inflammation. Furthermore, neutrophils play an under-appreciated role in regulating multiple aspects of immunity, including priming adaptive immunity and the resolution of immune responses (*Mayadas et al., 2014*). Neutrophils have been implicated in multiple non-infectious disease states, including autoimmune diseases, acute lung injury following trauma or sepsis, preeclampsia, atherosclerosis, and cancer (*Jorch and Kubes, 2017*).

Neutrophils arise from hematopoietic stem cells (HSPCs) resident within the bone marrow through a process that is regulated by the cytokine G-CSF (*Furze and Rankin, 2008*). Neutrophil differentiation begins with granulocyte/monocyte progenitor (GMP) cells that also give rise to monocytes, erythrocytes, eosinophils, and basophils. The first committed neutrophil progenitor is the neutrophil promyelocyte, which give rise to myelocytes, meta-myelocytes, banded neutrophils, and finally mature neutrophils (*Hidalgo et al., 2019*). Throughout the differentiation process, neutrophil progenitors gradually acquire the characteristics of mature neutrophils, including the production of primary/azurophilic, secondary/specific, and tertiary/gelatinase granules as well as the characteristic multi-lobular nucleus. Neutrophil differentiation is regulated by a series of transcription factors active at different stages of granulopoiesis (*Fiedler and Brunner, 2012*). Notably, deficiency in the transcription factor C/EBPε causes neutrophil-specific granule deficiency in humans, a rare disorder characterized by a lack of mature neutrophils and recurrent pyogenic infections (*Lekstrom-Himes et al., 1999*). Under normal homeostasis, an estimated $10^{11}$ mature neutrophils are released into circulation every day. These cells have a short half-life of ~6.5 hr up to ~5 days (*Furze and Rankin, 2008*; *Hidalgo et al., 2019*).

Mature neutrophils are capable of being recruited into tissues where they are further activated by inflammatory signals. Although it was previously believed that neutrophil activation resulted in a homogeneous population of terminally differentiated cells, recent discoveries have pointed to heterogeneity and plasticity in neutrophil function (*Mayadas et al., 2014*; *Rosales, 2018*). Advances in single-cell transcriptomics have demonstrated that even before activation, there are distinct subpopulations of neutrophils (*Paul et al., 2015*; *Xie et al., 2020*). Furthermore, numerous studies are now finding evidence of specialized neutrophil subpopulations in different disease states, though whether these populations represent distinct 'subsets' is unclear (*Silvestre-Roig et al., 2019*). New technologies for single-cell profiling have facilitated the identification of unique transcriptional subsets of neutrophils. However, our ability to probe the mechanisms of neutrophil differentiation and function has been limited by the short lifespan of mature neutrophils isolated ex vivo and the lack of appropriate cell lines that truly recapitulate neutrophil biology.

The ER-Hoxb8 method of conditionally immortalizing myeloid cells was developed to facilitate the generation of murine neutrophils and macrophages in large quantities from bone marrow HSCs (*Wang et al., 2006*). The method utilizes expression of a fusion of the estrogen receptor and the transcription factor Hoxb8, whose expression in the nucleus results in a block of terminal differentiation in cells cultured in the cytokine SCF (*Knoepfler et al., 2001*; *Owens and Hawley, 2002*). Estrogen binding to the ER-Hoxb8 construct results in translocation of Hoxb8 to the nucleus, while this construct will be sequestered in the cytosol in the absence of estrogen. ER-Hoxb8 neutrophil progenitors are immortalized in the GMP state and can be maintained in this immature state with the simple addition of estrogen, differentiating into mature neutrophils only after estrogen withdrawal (*Wang et al., 2006*). These differentiated neutrophils recapitulate many aspects of primary neutrophil function, including ROS production, NET formation, and killing of a bacterial pathogen (*Shannon and Hinnebusch, 2023*; *Wang et al., 2020*; *Wolach et al., 2021*). Thus, these cells offer the opportunity

to study the developmental transition from GMP to mature neutrophil, and to analyze genetic determinants of neutrophil function.

Neutrophils have long been known to be challenging to genetically manipulate due to their very short half-life in vitro. Early studies attempting to genetically manipulate neutrophils primarily relied on viral transduction of hematopoetic progenitor cells isolated from humans (*Lachmann et al., 2015*; *Sekhsaria et al., 1993*), a technique that is not easily scalable for genome-wide loss of function screening. The development of the CRISPR/Cas9 system of gene editing has opened the door to rapid editing of cells to create specific mutations of interest (*Jinek et al., 2012*; *Ran et al., 2013*; *Sanjana et al., 2014*). CRISPR pooled screens offer the chance to interrogate gene function in a large scale and unbiased fashion, and have been successful in both primary and immortalized cell lines (*Doench, 2018*; *Joung et al., 2017*; *Parnas et al., 2015*). In addition, several groups have demonstrated that Cas9/CRISPR can be used to edit hematopoetic progenitor cells, both in vivo and in vitro, with potential application to neutrophils (*Dong et al., 2021*; *LaFleur et al., 2019*). In addition, efficient editing of HoxB8 progenitors by nucleofection has been demonstrated and used to study the phenotypes of individual genes (*Khoyratty et al., 2021*; *Shannon and Hinnebusch, 2023*). We sought to further develop ER-Hoxb8 neutrophil progenitors as a system that is amenable to large-scale genetic screens. Using methods described to create the original ER-Hoxb8 neutrophil progenitor cell line, we created a new ER-Hoxb8 neutrophil progenitor line that expresses Cas9. Here, we show that these cells can be efficiently edited with Cas9 by introduction of a single guide RNA (sgRNA) using lentiviral transduction in the progenitor state. In addition, these cells acquire characteristics of mature neutrophils upon differentiation as expected, both in vitro and when transferred into mice in vivo. As a proof of concept of the utility of this cell line for screening, we performed a genome-wide survival-based screen to identify sgRNAs that prevent differentiation of neutrophils from progenitors. We demonstrate that Cas9-expressing ER-Hoxb8 neutrophils are a valuable system for studying GMP to mature neutrophil differentiation and function and suggest that this tool can be widely applied to study numerous aspects of neutrophil biology both in vitro and in vivo.

## Results

### Establishment of a Cas9-expressing ER-Hoxb8 neutrophil model

We first aimed to establish ER-Hoxb8 progenitor-derived neutrophils as a genetically tractable model for ex vivo neutrophils. Bone marrow hematopoietic stem cells (HSCs) were isolated from a C57BL/6 J mouse constitutively expressing Cas9 under the *Rosa26* genetic locus (*Platt et al., 2014*). HSCs were transduced with a murine stem cell virus (MSCV) vector encoding a fusion protein of the estrogen receptor and Hoxb8 transcription factor (ER-Hoxb8). Transduced HSCs were cultured in the presence of stem cell factor (SCF) cytokine and β-estradiol, leading to the production of immortalized progenitors committed to the neutrophil lineage (*Wang et al., 2006*). Upon withdrawal of estrogen (β-estradiol) Cas9[+]ER-Hoxb8 progenitor proliferation slowed and stopped by 4 days post-differentiation, at which time cells appeared smaller with many displaying condensed nuclear morphology, in line with published reports (data not shown) (*Wang et al., 2006*). To determine whether Cas9[+]ER-Hoxb8 cells express cell surface markers of neutrophil differentiation, we withdrew estrogen and differentiated progenitors for 3 or 4 days. Four-day differentiated Cas9[+]ER-Hoxb8 neutrophils expressed myeloid integrin CD11b and neutrophil surface antigen Gr-1, while the macrophage marker F4/80 was increased at 3 days but downregulated after 4 days of differentiation (*Figure 1A*). The cytokine G-CSF is known to specifically promote granulocyte differentiation and granulopoiesis (*Panopoulos and Watowich, 2008*) and has been used to induce neutrophil differentiation with several in vitro neutrophil cell models (*Gupta et al., 2014*). Multiple research groups have utilized ER-Hoxb8 neutrophil progenitors with G-CSF cytokine addition during estrogen withdrawal to augment neutrophil differentiation (*Chen et al., 2011*; *Gautam et al., 2013*; *Wang et al., 2006*). Similarly, we observed increased frequencies of CD11b+ cells that also expressed the neutrophil maturation marker Ly6G in cells cultured with G-CSF (*Figure 1B*). G-CSF addition during differentiation, therefore, appears to potently polarize ER-Hoxb8 neutrophil progenitors towards a more mature phenotype. To determine whether the differentiated cells exhibited morphological features characteristic of mature neutrophils, we performed transmission electron microscopy (TEM) on progenitor cells four days after withdrawal of estrogen. As expected, Cas9[+]ER-Hoxb8 progenitors were round cells with a large, unilobar nucleus

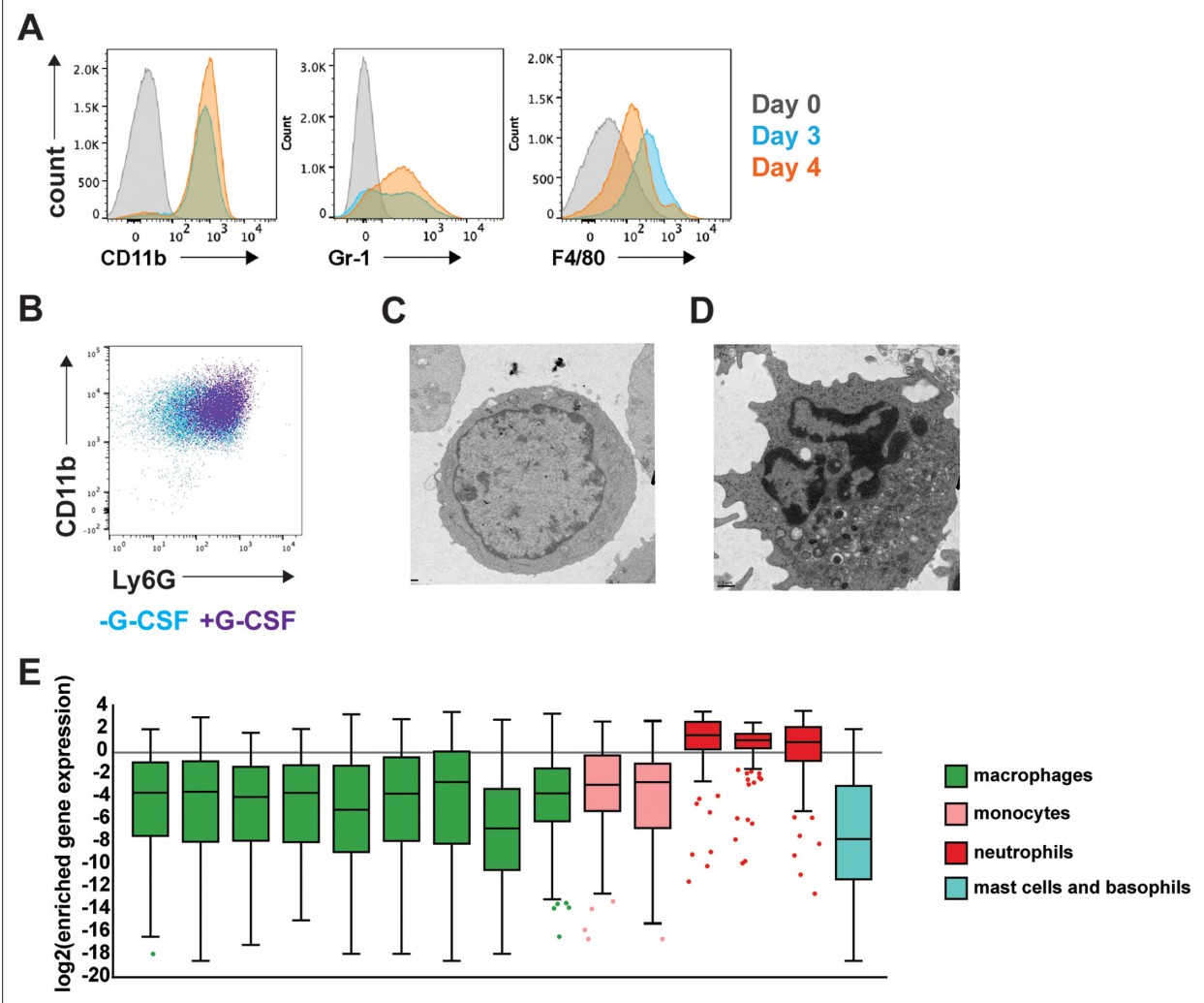

**Figure 1.** Differentiated Cas9+ER-Hoxb8 differentiated neutrophils closely resemble primary neutrophils. (**A**) Cas9+ER-Hoxb8 cells differentiated for 0, 3, and 4 days were analyzed for expression of indicated myeloid markers by flow cytometry. (**B**) Flow cytometry analysis of CD11b and Ly6G staining after 4 days of differentiation in the presence and absence of G-CSF. (**C, D**) Transmission electron microscopy (TEM) of Cas9+ER-Hoxb8 before (**C**) differentiation (**D**) and after 4 days of differentiation. (**E**) Comparison of gene expression profile of Cas9+ER-Hoxb8 after 2 days of differentiation with RNAseq data from the Immunology Genomes database. Y-axis represents log2(gene expression value/average expression value of all genes) for different cell types; in order from left to right: macrophages (MF_PC, MF_Fem_PC, MF_226+II + 480lo_PC, MF_RP_Sp, MF_Alv_Lu, MF_pIC_Alc_Lu, MF_microglia_CNS, MF_AT), monocytes (Mo_6C+II-_Bl, Mo_6C-II-), neutrophils (GN_BM, GN_Sp, GN_Thio_PC), and mast cells (MC_heparainase_PC). All data are representative of at least two experiments.

(*Figure 1C*). In contrast, differentiated Cas9+ER-Hoxb8 progenitor cells exhibited the neutrophil characteristics of an amorphous ameboid shape, a multi-lobed nucleus, and numerous granular structures evident in the cytosol (*Figure 1D*).

To better characterize these cells and more fully establish these cells as neutrophils, we performed RNAseq at day 0 prior to estrogen withdrawal and on day 2 post estrogen withdrawal in the presence of G-CSF. Comparison of gene expression profiles from these two timepoints performed in triplicate revealed >4000 genes that were differentially expressed (*Supplementary file 1*). Myeloid cells have overlapping transcriptional profiles and can be difficult to distinguish based on the expression of marker genes. Three genes found previously to be highly specific to neutrophils were dramatically upregulated after 2 days of differentiation in Cas9+ER-Hoxb8 cells: *Stfa2l1* (~166 fold), *Mrgpra2a* (~ sixfold), and *Mrgpra2b* (~ sixfold) (*Ericson et al., 2014*). We also compared the top 200 differentially expressed gene from our dataset to scRNAseq data from multiple macrophage, monocyte, neutrophil, and mast cell/basophil types from the Immunology Genomes database (https://www.

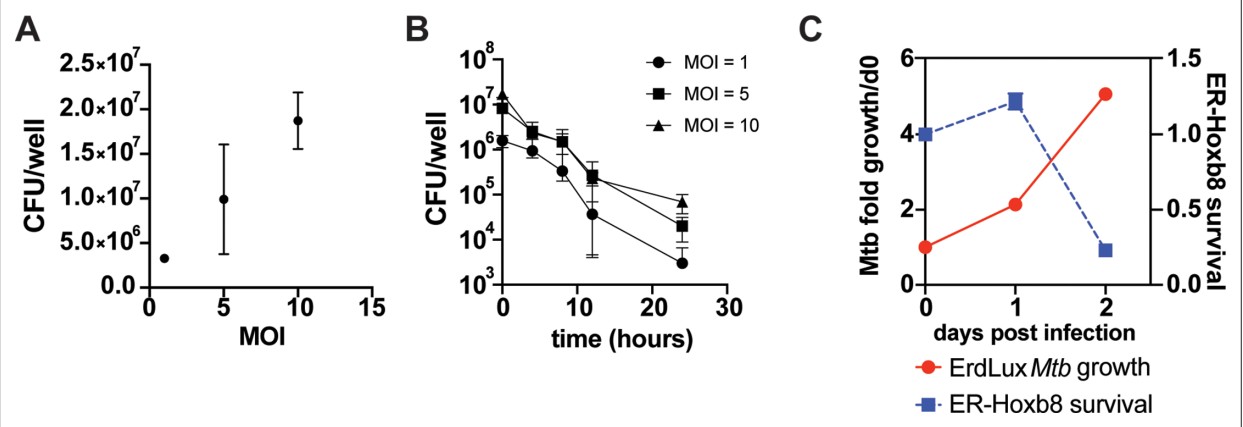

**Figure 2.** Differentiated Cas9⁺ER-Hoxb8 progenitors recapitulate primary neutrophil responses to methicillin resistant *Staphylococcus aureus* (MRSA) and *M. tuberculosis*. (**A**) Differentiated neutrophils were infected at the indicated multiplicities of infection (MOI) with MRSA USA300 and bacterial number (CFU) was measured immediately after phagocytosis. (**B**) Differentiated neutrophils effectively kill MRSA over a timecourse of infection as measured by CFU at indicated timepoints. (**C**) Differentiated neutrophil survival assayed using CellTiter Glo after infection with *M. tuberculosis* Erdman strain. Bacterial numbers were assayed using a luciferase (lux) reporter. All experiments were repeated a minimum of three times.

immgen.org/Databrowser19/DatabrowserPage.html). Importantly, the top 200 genes that were more highly expressed after 2 days of differentiation of Cas9⁺ER-Hoxb8 were enriched in neutrophil, but not other myeloid cell transcriptomes (*Figure 1E*). Taken together, these data rigorously establish differentiated Cas9⁺ER-Hoxb8 as a cell type that closely resembles primary neutrophils.

## Cas9⁺ER-Hoxb8 neutrophils recapitulate antimicrobial responses of primary neutrophils

We next sought to test whether Cas9⁺ER-Hoxb8 cells exhibit functional characteristics of mature neutrophils in the context of infection. Neutrophils are important for killing and eliminating the bacterial pathogen *Staphylococcus aureus* during infection, and patients with congenital neutropenia are susceptible to *S. aureus* infections (*Donadieu et al., 2011*). A previous study demonstrated Hoxb8 neutrophils are capable of phagocytosing *S. aureus*, however, their ability to kill pathogens has not been evaluated. We infected Cas9⁺ER-Hoxb8 neutrophils with the USA300 strain of methicillin resistant *Staphylococcus aureus* (MRSA) at three different multiplicities of infection (MOI), enumerated bacteria after phagocytosis and found that the bacteria were effectively phagocytosed in a manner linear with input (*Figure 2A*). Numbers recovered at the t=0 timepoint were higher than input, perhaps due to bacterial replication during the 1 h phagocytosis period. Differentiated Cas9⁺ER-Hoxb8 neutrophils effectively killed MRSA, decreasing bacterial numbers from 2 to 3 logs over a 24 hr period, demonstrating that these cells have potent bactericidal capacity (*Figure 2B*). In contrast to the clear role of neutrophils in controlling *Staph* infection, neutrophils play a detrimental role in infection with the bacterial pathogen *Mycobacterium tuberculosis*. Neutrophils are unable to kill *M. tuberculosis*, and in fact contribute to host susceptibility. Several reports have demonstrated that neutrophils phagocytose *M. tuberculosis* but die rapidly upon infection (*Alemán et al., 2004*; *Dallenga et al., 2017*). In contrast to what we observed with MRSA, we found that *M. tuberculosis* replicated within Cas9⁺ER-Hoxb8 neutrophils during the first day after infection, a period of time during which neutrophils remained viable (*Figure 2C*). By day 2 after infection, infected neutrophils were killed, suggesting that this cell line also recapitulates responses to infection previously reported for primary mouse and human neutrophils (*Figure 2C*). Taken together, these results provide evidence that Cas9⁺ER-Hoxb8 neutrophils recapitulate interactions with two important bacterial pathogens.

## Efficient gene editing in Cas9⁺ER-Hoxb8 neutrophils

Gene editing capacity via Cas9 expression is a potentially important benefit of ER-Hoxb8 neutrophil progenitors as tractable models, as ex vivo neutrophils are not easily genetically modified (*Saul et al., 2019*). To test the gene knockout efficiency using CRISPR/Cas9-mediated gene editing, two sgRNA sequences per gene selected from the optimized Brie library (*Doench et al., 2016*) or a control

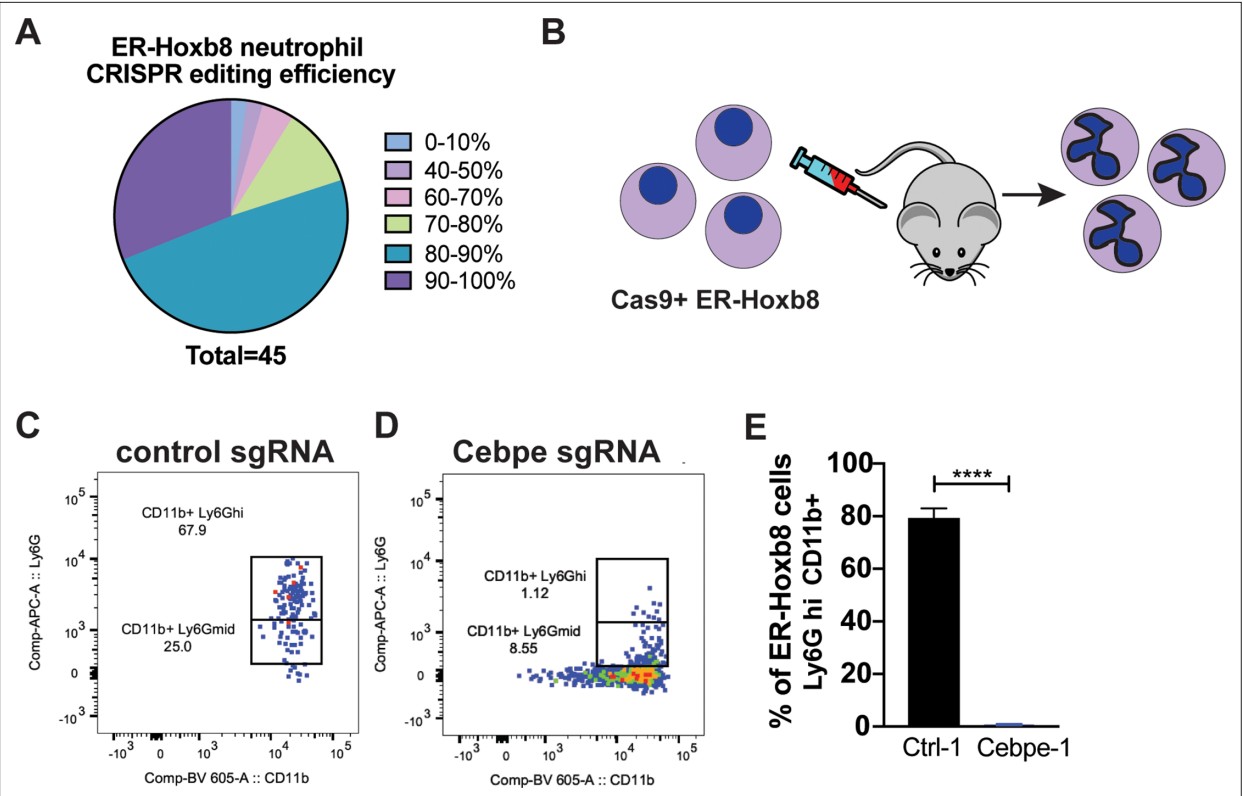

**Figure 3.** Efficient CRISPR/Cas9 editing in differentiated Cas9+ER-Hoxb8 cells in vitro and in vivo. (**A**) Editing efficiency of individual single guide RNAs (sgRNAs) was analyzed using TIDE analysis. (**B**) Edited Cas9+ ER-Hoxb8 progenitor cells were transferred into irradiated mice, and differentiation was analyzed at 5 days post transfer. (**C, D, E**) Flow cytometry analysis of in vivo differentiation of neutrophil progenitors at 5 days post transfer into irradiated recipients using progenitors edited with a control guide (**C, E**) or with a sgRNA targeting *Cebpe* (**D, E**). Neutrophils are identified as Ly6Ghi and CD11b positive. All experiments were repeated 2–3 times. Unpaired *t*-test *p*-value calculated in comparison with nontargeting control cell line. ****$p < 0.0001$.

The online version of this article includes the following figure supplement(s) for figure 3:

**Figure supplement 1.** Time course of differentiation of Hoxb8+ neutrophil progenitors in vivo.

nontargeting sgRNA were cloned into the lentiGuide-Puro expression vector (*Sanjana et al., 2014*). These constructs were then packaged in lentivirus and transduced into Cas9+ER-Hoxb8 neutrophil progenitors. After antibiotic selection for stably transduced cells, PCR-amplified genomic DNA (gDNA) from targeted knockout cell lines was sequenced and aligned to amplicons from a control nontargeted cell line. TIDE analysis (*Brinkman et al., 2014*) was used to estimate editing efficiency. Editing efficiency was high, with >25% of guides achieving >90% editing efficiency, and more than 75% achieving higher than 80% efficacy (*Figure 3A*). These data indicate that efficient CRISPR editing is possible using Cas9+ER-Hoxb8 neutrophil progenitors.

It was recently demonstrated that ER-Hoxb8 neutrophil progenitors effectively differentiate into neutrophils when transferred into irradiated recipient mice (*Orosz et al., 2021*). We next set out to test whether we observe differentiation and Cas9-dependent editing of Cas9+ER-Hoxb8 neutrophil progenitors in vivo. We first transferred ER-Hoxb8 neutrophil progenitors transduced with a control sgRNA into irradiated mice and analyzed neutrophils in the blood after 5 days (*Figure 3B*). These cells expressed high levels of both Ly6G and CD11b, indicating they effectively differentiated into neutrophils in vivo (*Figure 3C*). C/EBPε is a transcription factor known to be essential for neutrophil differentiation during the promyelocyte to metamyelocyte stage (*Lawrence et al., 2018*). We transduced Cas9+ER-Hoxb8 neutrophil progenitors with a lentiviral sgRNA construct targeting *Cebpe* or a non-targeting sgRNA and transferred these cells into irradiated recipient mice. At five days after transfer, we observed far fewer CD11b+ Ly6Ghi mature neutrophils in mice that received *Cebpe*-targeted cells (*Figure 3D and E*). To demonstrate that this was not a result simply of a delay in the

emergence of neutrophils, we performed a kinetic analysis of control and *Cepbe*-targeted cells after transfer. Neutrophils derived from control sgRNA transduced progenitors rose in numbers over time, peaking at day 5–6 (*Figure 3—figure supplement 1*). This is consistent with previous observations of the differentiation of ER-Hoxb8 neutrophils in vivo (*Orosz et al., 2021*). At no timepoint did the percentage of CD11b$^+$ Ly6G$^{hi}$ *Cepbe* targeted cells reach the levels of wild-type Cas9$^+$ER-Hoxb8 progenitors, confirming that this transcription factor is required for differentiation in vivo and that this phenotype can be recapitulated with Cas9$^+$ER-Hoxb8 progenitors.

## Screen for genetic modulators of neutrophil survival during differentiation

Pooled CRISPR sgRNA libraries have been widely used for genome-wide screening of cellular phenotypes. Previously, this technology has not been applicable to neutrophils. We set out to create a genome-wide sgRNA library in Cas9$^+$ER-Hoxb8 neutrophil progenitors. We used the Brie gRNA library of 78,637 mouse sgRNAs, which was designed to have an average of 4 sgRNAs—optimized for off-target effects—per gene, targeting 19,674 genes (*Doench et al., 2016*). 1000 control non-targeting sgRNAs were also included for normalization. Neutrophil progenitors were transduced with lentivirus containing the mouse CRISPR lentiviral pooled knockout Brie library on the lentiGuide-Puro backbone at an MOI of 0.11. Transduction conditions were optimized such that each transduced cell would have a single sgRNA expressed. After transduction, cells were treated with antibiotics to select for stable transductants. 500 x coverage minimum of all sgRNAs was maintained at all times during transduction, selection, and passaging to avoid losing sgRNA representation or biasing the screen (*Doench et al., 2016*; *Joung et al., 2017*).

To characterize the performance of library for genome-wide screening, and to demonstrate the utility of the cells for addressing a biologically significant question, we performed a screen to identify genes that modulate the differentiation of Cas9$^+$ER-Hoxb8 progenitors into mature neutrophils. To carry out the screen, we differentiated 40 million Cas9$^+$ER-Hoxb8 progenitors by β-estradiol withdrawal in the presence of G-CSF. In parallel to the differentiated neutrophils, 40 million progenitors were maintained in the presence of β-estradiol to serve as an undifferentiated control. At the end of 4 days of passaging in culture, gDNA was harvested from the undifferentiated and differentiated conditions, guide RNA sequences were PCR amplified, and Illumina sequencing adapters were ligated. The libraries were then sequenced on an Illumina HiSeq 4000 to detect which sgRNAs led to reduced or enhanced survival during differentiation (*Figure 4A*). Analysis using the MAGeCK algorithm (*Li et al., 2014*) allowed for ranking of sgRNA scores using a robust rank algorithm (RRA) for negative selection in the differentiated cell pool (*Figure 4B*). The control sgRNAs appeared to be widely distributed on a plot of all sgRNA counts, as expected (*Figure 4—figure supplement 1A*). However, non-targeting sgRNAs tended to have higher numbers of sgRNA counts than targeting sgRNAs (*Figure 4—figure supplement 1B*). Targeting sgRNAs had a distribution narrower than that of a normal distribution and many sgRNAs with fewer than 50 counts (*Figure 4—figure supplement 1B*), which may be indicative of a tendency for targeting sgRNAs to induce deleterious effects leading to poorer cell survival.

MAGeCK RRA scores were used to rank genes based on negative selection, implicating genes that may be necessary for neutrophil survival during differentiation, and also for genes that resulted in increased survival during differentiation (*Supplementary file 2*). To identify pathways and protein complexes required for neutrophil survival during differentiation, we used MAGeCKFlute to perform gene set enrichment analysis using Gene Ontology Cellular Component terms (*Wang et al., 2019*). The top ten enriched categories included the protein complexes GATOR2 and KICSTOR, regulators of mTORC signaling, as well as the mitochondrial pyruvate dehydrogenase complex (PDC) (*Figure 4C*). In addition, the WASH complex, a type I nucleation promoting factor that activates the Arp2/3 complex and is essential for endosomal membrane scission and cargo sorting, emerged as a major class of hits from the negative selection screen (*Figure 4C*; *Duleh and Welch, 2010*; *Linardopoulou et al., 2007*).

We next sought to determine whether screening for survival during Cas9$^+$ER-Hoxb8 differentiation can identify biologically meaningful genes required for neutrophil differentiation or survival. To do so, we first set out to demonstrate that the screen detected genes known to be required neutrophil differentiation or survival, and to validate the results of the screen by re-testing hits in individual sgRNA targeting experiments. Importantly, one of the top hits from the screen was the gene *Cebpe*. In addition, hits included the gene *Ep300* that encodes the acetylase p300, a regulator of C/EBPε

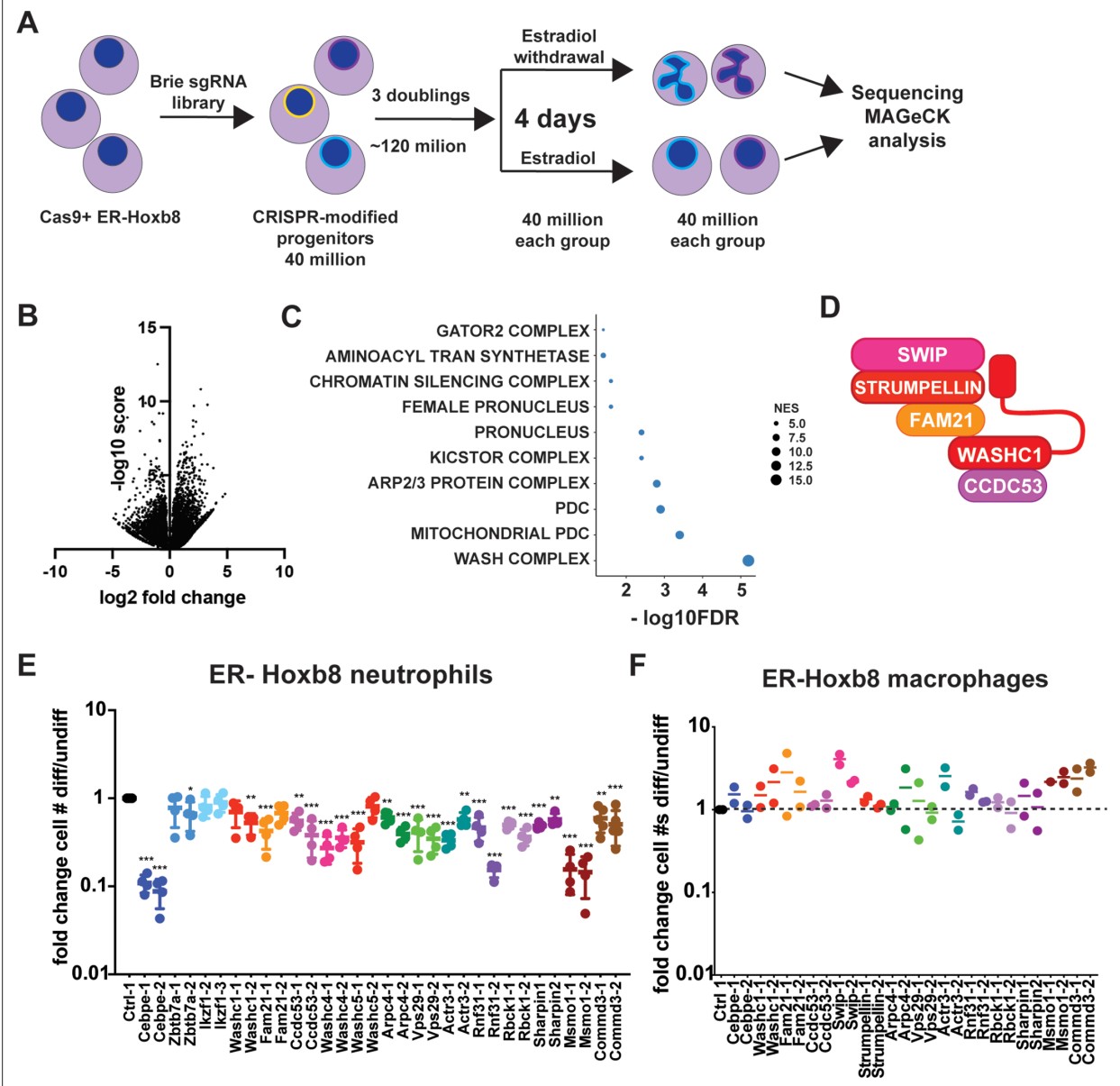

**Figure 4.** CRISPR screen using Cas9+ER-Hoxb8 library reveals factors required for survival during neutrophil, but not macrophage, differentiation. (**A**) Schematic of the CRISPR screen. Cas9+HoxB8 library was created. Screening is initiated by growing up cells for a minimum of 500 x coverage. These cells are differentiated by estradiol withdrawal for 4 days, at which time DNA is prepared and sequenced (**B**) Volcano plots of results from the CRISPR screen. (**C**) GSEA using MAGeCK Flute analysis depicts GO cellular compartment categories negatively enriched in the differentiation screen. (**D**) Members of the WASH complex. (**E, F**) Survival of individual Cas9+ER-Hoxb8 progenitor cell lines targeted with individual single guide RNA (sgRNA) selected from the Brie library was determined prior to differentiation and at 4 days post-differentiation. The same sgRNAs were used to target both neutrophil (**E**) and macrophage (**F**) progenitors. All experiments were repeated a minimum of two times. The CRISPR screen was performed in biological triplicate. *$p<0.05$, **$p<0.01$, ***$p<0.001$ by one-way ANOVA with multiple comparisons.

The online version of this article includes the following figure supplement(s) for figure 4:

**Figure supplement 1.** CRISPR pooled screen single guide RNA (sgRNA) read counts.

activity (*Bartels et al., 2015*) and is, therefore, another important positive control as an essential factor in neutrophil differentiation. Genes encoding all five members of the WASH complex were also identified as hits (*Figure 4D*). In addition to the core WASH complex, we identified three endosomal trafficking-associated proteins that are part of the Arp2/3 complex (*Arpc4*, *Actr3*) or retromer complex (*Vps29*), which have been implicated in WASH function (*Alekhina et al., 2017*). The WASH complex was previously shown to be required for the maintenance of hematopoiesis, and adult mice

with induced WASH deletion in all cells have severely depleted numbers of short-term hematopoietic stem cells (ST-HSCs) (*Xia et al., 2014*). However, our screening data suggest that WASH may also be specifically involved downstream in neutrophil differentiation. Finally, *Zbtb7a* and Ikzf1 were at the top of the list of negatively selected hits in the CRISPR differentiation screen (*Supplementary file 2*). Both genes have been implicated in lymphocyte and in some cases myeloid differentiation (*Boutboul et al., 2018*; *Constantinou et al., 2019*), however, they have not previously been implicated in neutrophil differentiation or survival.

We selected a set of 16 genes identified as hits in the screen for validation in individual sgRNA targeting experiments (*Figure 4E*). These genes included the positive control *Cebpe*, all WASH complex component genes (*Wash1*/WASHC1, *Fam21*/FAM21, *Ccdc53*/CCDC53, *A230046K03Rik/ Washc4*/SWIP, and *E430025E21Rik*/STRUMPELLIN), as well as genes encoding factors involved in endosomal trafficking (*Arpc4, Vps29, and Actr3*), linear ubiquitination (LUBAC) (*Rnf31, Rbck1, Sharpin*), cholesterol metabolism (*Msmo1*), and membrane trafficking and transcription regulation (*Commd3*). We selected 2 sgRNAs per gene and used lentiviral transduction to target these genes for deletion in Cas9$^+$ER-Hoxb8 neutrophil progenitors. Differentiation into neutrophils was induced in sgRNA-targeted progenitors by withdrawal of β-estradiol in the presence of G-CSF, and cell numbers were counted at 4 days post differentiation in both differentiated and undifferentiated cells (*Figure 4E*).

With the exception of *Zbtb7a* and *Ikzf1*, all the tested hits from the screen recapitulated the expected phenotype of a defect in survival during differentiation into neutrophils, but not in progenitor cells (*Figure 4E*). To test whether the observed phenotypes were specific to neutrophils, we turned to a recently developed Cas9$^+$ER-Hoxb8 immortalized cell line that differentiates into macrophages from the progenitor state (CIMs) (*Roberts et al., 2019*). We used the same sgRNA constructs to target genes in macrophage progenitors and differentiated these cells into macrophages by withdrawal of β-estradiol in the presence of GM-CSF. Interestingly, targeting genes implicated in neutrophil survival or differentiation had no impact on survival of CIMs during differentiation (*Figure 4F*), suggesting that the functions of these genes are neutrophil-specific. These results demonstrate that Cas9ER-Hoxb8 immortalized neutrophil progenitor libraries can be used to successfully identify genes required for neutrophil survival and differentiation. Taken together, our results also suggest that CRISPR screens using this tool can be performed to interrogate other aspects of neutrophil biology.

## Discussion

Neutrophils play an essential role in defense against infection. Dysregulation of neutrophil function contributes to numerous disease states, both infectious and also non-communicable. Understanding neutrophil function has been hampered by the difficulty of working with these cells, as primary neutrophils die rapidly upon isolation. The Hoxb8 immortalized neutrophil cell line was a significant advance in that it enables the production of large numbers of neutrophils in vitro that terminally differentiate from neutrophil progenitors. Expression of an estrogen-dependent fusion protein with Hoxb8 allows for conditional immortalization in the presence of β-estradiol; withdrawal of β-estradiol results in differentiation of neutrophils. Here, we describe an adaptation of this cell line in which we have isolated a new Cas9-expressing Hoxb8 immortalized neutrophil cell line from Cas9 transgenic mice. This resource enables both forward and reverse genetics in cells that recapitulate numerous aspects of neutrophil biology, which will facilitate mechanistic investigations into neutrophil function both in vitro and in vivo.

We demonstrated that Cas9$^+$ER-Hoxb8 progenitors differentiate into cells that resemble primary neutrophils upon withdrawal of β-estradiol both morphologically and functionally. Other groups have demonstrated ER-Hoxb8 progenitors resemble mature neutrophils upon differentiation and demonstrate conventional neutrophil functions, including producing a respiratory burst and performing phagocytosis. We demonstrate that Cas9$^+$ER-Hoxb8 derived neutrophils also morphologically resemble mature primary neutrophils. Moreover, we demonstrate that they exhibit expected phenotypes during bacterial infection: while they effectively kill methicillin-resistant *S. aureus*, they are unable to kill *M. tuberculosis* and die rapidly upon infection. In addition, HoxB8 neutrophil cells that were edited using delivery of Cas9 using nucleofection were also shown to be useful in studies of *Yersinia pestis* pathogenesis (*Shannon and Hinnebusch, 2023*). Thus, these cells will be useful for mechanistic investigations of neutrophil function during bacterial infection.

Expression of Cas9 in the progenitor cells facilitated effective genome editing that was maintained when progenitors were differentiated into neutrophils. Editing was highly efficient, with more than 75% of sgRNAs tested resulting in >80% editing efficiency in differentiated neutrophils. Furthermore, we demonstrated that edited progenitor cells can be used to study neutrophil function in vivo. C/EBPε is a transcription factor known to be important for neutrophil differentiation. We transduced Cas9⁺ER-Hoxb8 progenitors with lentiviral constructs encoding 2 individual sgRNA constructs targeting *Cebpe*. Similar to what has been reported for non-Cas9-expressing Hoxb8 progenitor cells, Cas9⁺ER-Hoxb8 progenitors effectively differentiated into neutrophils in vivo when transferred into irradiated mice. *Cebpe*-targeted progenitors exhibited a defect in neutrophil differentiation in vivo, demonstrating the expected phenotype based on the known function of this transcription factor and validating that this approach can be used to study neutrophil gene function in vivo. Although we only used one transfer of progenitors to test the effect of mutation *Cebpe,* repeated transfer of cells results in a sustained level of neutrophils derived from ER-Hoxb8 progenitors (*Orosz et al., 2021*), suggesting that this technique can be used to study neutrophil function over longer periods of time.

To validate that Cas9⁺ER-Hoxb8 progenitors can be used for forward genetic screens, we performed a pooled screen to identify genes required for neutrophil differentiation or survival after differentiation. We identified hundreds of genes that were either positively or negatively selected for during 4 days of differentiation into neutrophils. Importantly, one of the top hits from the screen was *Cebpe,* a positive control that validates that this screen can successfully identify factors involved in neutrophil differentiation. By retesting hits in both Cas9⁺ER-Hoxb8 neutrophil progenitors and Cas9⁺ER-Hoxb8 macrophage progenitors, we were able to demonstrate that hits were specific to survival during neutrophil, and not macrophage, differentiation from the myeloid progenitor state. The screen was highly sensitive, identifying every member of the WASH complex as being required for neutrophil survival during the differentiation period. WASH was previously found to be required for hematopoiesis, perturbing the differentiation of long-term hematopoietic stem cells into subsequent progenitors (*Xia et al., 2014*). Our data suggest that the WASH complex is also involved more specifically at later stages of neutrophil development. In addition, we identified hits in other pathways and biological processes, including endosomal dynamics, linear ubiquitin, and cholesterol biosynthesis. It is interesting that we identified three components of LUBAC as being required for survival during neutrophil, but not macrophage differentiation. It was previously shown that neutrophils from mice lacking *Sharpin* were highly sensitive to cell death induced by TNF (*Rickard et al., 2014*). It is possible that neutrophils lacking components of LUBAC are sensitized towards death during the neutrophil differentiation process or die rapidly upon differentiation.

It is important to note, however, that there are some limitations of this study. First, while we and others have found that HoxB8 immortalized neutrophils are phenotypically highly similar to primary neutrophils, most of this characterization has occurred in vitro. It remains to be seen whether these neutrophils similarly recapitulate neutrophil function in vivo. Nonetheless, we have shown that HoxB8 immortalized neutrophils can be used to interrogate the function of specific genes in vivo. It remains to be determined whether HoxB8 Cas9 + neutrophils can be effectively used for in vivo screening. Second, this study primarily presents a technical resource that demonstrates the feasibility of using HoxB8 Cas9 + neutrophils to identify novel regulators of neutrophil differentiation and survival. Further investigation is necessary to clearly establish a biological/mechanistic role for genes we identified in our screens for neutrophil differentiation, both in vivo and in vitro. Future work on the hits described could lead to new tools for manipulating neutrophil numbers in vivo as and for understanding the pathways that regulate their survival.

Broadly speaking, genome-wide screens for neutrophil function have been challenging, even for relatively established phenotypes, such as bacterial killing. This resource may now enable such screens that could identify genes required for neutrophil antimicrobial activity. Recent genome-wide screens for neutrophil function using the HL-60 cell line *Belliveau et al., 2023*; *Nagy et al., 2024* have provided insights into neutrophil migration. However, whether HL-60 cells fully recapitulate primary neutrophil function is uncertain (*Collins, 1987*; *Gupta et al., 2014*). HoxB8 progenitors may facilitate genome-wide screens for a wider array of neutrophil functions. For some phenotypes, alternative strategies—such as CRISPRi libraries or those with higher guide RNA coverage—may offer advantages. Finally, for detailed interrogation of individual genes, generating transgenic mice or directly editing bone marrow progenitors may provide complementary benefits (*LaFleur et al., 2019*). Nonetheless,

the Cas9⁺ER-Hoxb8 system represents a valuable tool for interrogating neutrophil biology both in vitro and in vivo.

## Materials and methods

### Cas9⁺ER-Hoxb8 progenitor cell line generation

Cas9⁺ER-Hoxb8 (ER-Hoxb8) neutrophil progenitors were generated as previously described (*Wang et al., 2006*) using *Rosa26*-Cas9 knock-in B6J mice constitutively expressing Cas9 (Jackson Laboratory strain #026179) (*Platt et al., 2014*). All mice were housed in specific pathogen-free conditions and treated using procedures described in animal care protocols approved by the Institutional Animal Care and Use Committee of UC Berkeley. Immortalized neutrophil progenitors were cultured in ER-Hoxb8 neutrophil medium consisting of Opti-MEM (Gibco) supplemented with 10% FBS, 1% SCF supernatant produced by a CHO cell line, 2 mM GlutaMax supplement (Gibco), 30 μM β-mercaptoethanol, and 1 μM β-estradiol (Sigma). Neutrophil progenitors were passaged every 2–3 days in non-treated flasks to keep cell concentrations below $2\times10^6$ cells/mL. Progenitor lines were maintained in a 37 °C humidified incubator with 5% $CO_2$. Progenitors were cryopreserved in liquid nitrogen resuspended in 90% FBS and 10% DMSO. Cas9⁺ER-Hoxb8 macrophage progenitors (CIMs) were obtained from the Cox lab at UC Berkeley (*Roberts et al., 2019*). Undifferentiated CIMs were maintained in RPMI (Gibco) supplemented with 10% FBS, 2% GM-CSF supernatant produced by a B16 murine melanoma cell line, 2 mM L-glutamine, 1 mM sodium pyruvate, 10 mM HEPES, 43 μM β-mercaptoethanol, and 2 μM β-estradiol. Macrophage progenitors were passaged every 2–3 days in non-treated flasks to keep cell concentrations below $0.5\times10^6$ cells/mL. Cells tested negative for mycoplasma.

### ER-Hoxb8 progenitor differentiation

To differentiate neutrophil progenitors, cells were washed twice in sterile PBS, pH 7.4 (Gibco) to remove β-estradiol, then replated onto non-treated flasks in ER-Hoxb8 neutrophil medium without β-estradiol. Cell media was refreshed by pipetting cells up and down to resuspend and detach loosely adherent cells, then replating cells at a concentration sufficient to keep cell concentrations below $2\times10^6$ cells/mL. For G-CSF differentiation, 5 ng/mL murine G-CSF (Peprotech) was added to media on the day of β-estradiol withdrawal. After differentiating for 3 or 4 days, neutrophils were harvested from either only the floating fraction, or the floating fraction pooled with adherent cells harvested by incubating with sterile PBS at 4 °C for 15–20 min before pipetting to resuspend. Cells were either analyzed for surface marker expression, or replated onto treated plates for infections to be performed on day 4 of differentiation. For growth analysis, ER-Hoxb8 neutrophils were counted on day 0 of differentiation before plating, then replated on non-treated 6-well or 24-well plates. Cells were split on day 2 of differentiation; all floating cells from differentiated cells were split into new wells and media refreshed on both wells. Cells were counted by dilution in 0.4% trypan blue (VWR) on a hemacytometer. CIMs were differentiated similarly into CIM medium without β-estradiol for 9 days for growth analysis. For CRISPR screen single targeted cell line validation, cells were counted on the final day of differentiation and the fold change in cell counts compared to undifferentiated was calculated. Fold change values were normalized to that of the control nontargeted cell line.

### Flow cytometry analysis of myeloid markers

$1–10 \times10^6$ ER-Hoxb8 cells were harvested, washed in PBS, then stained with fixable viability dye eFluor 780 (eBioscience) for 20–30 min at 4 °C. Cells were washed in FACS buffer (PBS, 1% FBS, 2 mM EDTA), stained with Fc receptor block (BioLegend) for 10 min at 4 °C, then surface stained with antibodies against CD11b (M1/70, BioLegend), F4/80 (BM8, BioLegend), Gr-1 (RB6-8C5, BioLegend), and Ly6G (1A8, BioLegend) for 20–30 min at 4 °C. For in vivo differentiation, cells were also surface-stained with antibodies against CD45 (30F11, BioLegend), CD117 (2B8, BD), CD45.1 (A20, BD), CD3 (17A2, BioLegend), and B220 (RA3-6B2, BioLegend). Data were collected using a BD LSR Fortessa flow cytometer with FACSDiva software (BD Biosciences) at the UC Berkeley Cancer Research Laboratory Flow Cytometry Core and analyzed using FlowJo Software (Tree Star). ER-Hoxb8 cells were gated on live CD45⁺ CD45.1⁻ GFP⁺ (ER-Hoxb8⁺) singlets and analyzed for myeloid marker expression.

### Transmission electron microscopy

ER-Hoxb8 neutrophils were maintained in an undifferentiated state with β-estradiol or differentiated for four days with G-CSF. Cells were fixed with 2.5% glutaraldehyde/formaldehyde in 0.1 M sodium

cacodylate buffer, pH 7.2 (EMS) for at least 30 min prior to being stabilized in 1% low melting point agarose (EMS). Agarose was diced into 0.5 mm cubes, then fixed overnight. Samples were rinsed (3x; 10 min, RT) in 0.1 M sodium cacodylate buffer, pH 7.2, and immersed in a solution of 1% osmium tetroxide with 1.6% potassium ferricyanide in 0.1 M sodium cacodylate buffer for 1 hr. Samples were rinsed (3x; 10 min, RT) in 0.1 M sodium cacodylate buffer, pH 7.2; then subjected to an ascending acetone gradient (10 min; 35%, 50%, 70%, 80%, 90%, 100%) followed by pure acetone (2x; 10 min, RT). Samples were progressively infiltrated while rocking with Epon 812 resin (EMS) and polymerized at 60 °C for 24–48 hr. Thin sections (90 nm) were cut using a Leica UC6 ultramicrotome (Leica). Sections were then collected onto formvar-coated copper 50 mesh grids. The grids were post-stained with 2% uranyl acetate followed by Reynold's lead citrate, for 5 min each. The sections were imaged using an FEI Tecnai 12 120kV TEM (FEI) and data recorded using a Gatan Rio16 CMOS camera with GWS software (Gatan Inc).

## MRSA infections

The night before infection, a 25 mL LB culture was started from glycerol stock of WT MRSA USA300 SF8300 and shaken at 37 °C overnight. Following differentiation, ER-Hoxb8 neutrophils were plated on tissue-culture treated plates just prior to infection. The day of infection, the overnight MRSA culture was back diluted 1:60 into fresh 25 mL LB and shaken at 37 °C for 2 hr. Once the back dilution culture had reached mid-log (OD600~0.5), MRSA was washed twice with PBS by centrifuging at 1600×$g$ for 15 min per wash. Once the MRSA was resuspended at desired MOIs, ER-Hoxb8 neutrophils were spinfected with media containing MRSA at 350×$g$ for 5 min. After spinfection, the infected ER-Hoxb8 neutrophils were incubated at 37 °C 5% CO2 for 1 hr. After 1 hr incubation, the infected ER-Hoxb8 neutrophils were washed twice with PBS. For the t=0 hr timepoint, the ER-Hoxb8 neutrophils were lysed with autoclaved MilliQ water. MRSA was plated onto LB agar plates and incubated at 37 °C for ~14 hr. For other timepoints, ER-Hoxb8 neutrophils were resuspended with media containing 100 μg/mL gentamicin (Sigma-Aldrich) as a gentamicin protection assay and incubated at 37 °C, 5% CO$_2$ for the indicated amount of time. Once the timepoint was reached, the infected ER-Hoxb8 neutrophils were washed twice with PBS and lysed with autoclaved MilliQ water, followed by the same plating procedure as the t=0 hr timepoint.

## *M. tuberculosis* infection

Differentiated neutrophils were plated in 96-well tissue culture-treated black-walled plates at a density of 2.5×10$^5$ cells/well on day 4 of differentiation, immediately before infection. Cells were plated in triplicate for CellTiter Glo and in quadruplicate for luminescent bacterial growth assays. Erdman-lux (ErdLux) luminescent bacteria containing the luxBCADE operon (*Braverman et al., 2016*; *Penn et al., 2018*) were grown in Middlebrook 7H9 liquid media supplemented with 10% albumin-dextrose-saline, 0.4% glycerol, and 0.05% Tween 80 to an OD600 of 0.2–1 on the day of infection. Log phase *M. tuberculosis* bacteria were centrifuged at 2850×$g$ for 5 min, washed twice in PBS, sonicated three times for 30 s in a water bath sonicator, centrifuged at 60 x $g$ for 5 min to remove bacterial clumps, and resuspended in PBS to determine bacterial concentration. Bacteria were resuspended in horse serum for addition to neutrophils in progenitor medium at a final concentration of 5% horse serum and MOI of 1. Cells were maintained in a 37 °C humidified incubator with 5% CO$_2$.

Each day after infection, a half media change was performed after luminescent readings. For ErdLux experiments, plates were equilibrated for at least 10 min at 30 °C before luminescent readings were taken using a SpectraMax L plate reader. For cell viability/survival assays, plates were equilibrated at room temperature for at least 30 min before 100 μL of 1:1 diluted PBS and CellTiter Glo reagent (Promega) was added to 100 μL of media. Reactions proceeded for a minimum of 12 min while equilibrating at 30 °C before luminescent readings. All work with live *M. tuberculosis* was performed in the UC Berkeley Biosafety Level 3 (BSL3) facility.

## CRISPR/Cas9-mediated genetic editing

Guide RNA (gRNA) sequences either targeting mouse genes or control non-targeting single sgRNA sequences were taken from the Brie library of sgRNAs (*Doench et al., 2016*). Mouse Brie CRISPR knockout pooled library was a gift from David Root and John Doench (Addgene #73633). The plasmid DNA pool was amplified as previously described (*Doench et al., 2016*) at the UC Berkeley High

Throughput Screening Facility and sequenced to confirm guide representation at the UC Berkeley Vincent J. Coates Genomics Sequencing Laboratory on an Illumina HiSeq 4000, 50 SR with 10% PhiX, single 8 bp index. Single guide RNA oligonucleotides were synthesized by IDT and cloned into the lentiGuide-Puro (pLGP) plasmid vector following the LGP cloning protocol (*Sanjana et al., 2014*). Successful cloning was validated by isolating plasmid DNA from transformed *E. coli* Stbl3 cells using the QiaPREP Spin Miniprep kit, and Sanger sequencing using the hU6 primer at the UC Berkeley DNA Sequencing Facility. Sequences were aligned using Benchling and ApE software. Lentivirus containing sgRNA-pLGP was generated by transfecting Lenti-X 293T cells (Takara Bio) with sgRNA-pLGP, psPAX2, and pMD2 using Lipofectamine 2000 (Invitrogen) in Opti-MEM according to the manufacturer's protocols. Lentivirus-containing supernatants was harvested 48 hr post-transfection and filtered through a 0.45 μm syringe onto ER-Hoxb8 progenitors. Progenitors were spinfected in 6-well plates at 23–32°C for 30 min at 1000 *x g* with 4 μg/mL polybrene (Sigma) and 1 μM β-estradiol at $0.25 \times 10^6$ cells/mL for single gene knockouts. $5 \times 10^6$ cells/mL were transduced in non-treated 24-well plates at 23–32°C for 2 hr at 1000 *x g* with 4 μg/mL polybrene and 1 μM β-estradiol for Brie library pooled screen. Cells were diluted in progenitor media containing β-estradiol after spinfection. 48 hr after infection, cells were selected with 5 μg/mL puromycin for at least two days to select for successful transductants, at which point all cells in a non-transduced progenitor population would be selected against as a control. After transduction, cells recovered in ER-Hoxb8 medium without puromycin for several days before freezing. Frozen cells were thawed into progenitor medium with β-estradiol and passaged for several days to allow cells to recover before use in experiments. For TIDE analysis, PCR primers were designed for each sgRNA to amplify a stretch of DNA ~700 bp enclosing the designed editing site with the projected break site ~200 bp downstream from the sequencing start site. Genomic DNA from the targeted cell line and a control nontargeted cell line were used as primers. PCR amplicons were Sanger sequenced using one of the PCR primers at the UC Berkeley DNA Sequencing Facility and Elim Biopharm.

## In vivo adoptive transfer

Male congenic mice carrying the CD45.1 allele on the C57BL/6 J background (B6.SJL-Ptprc[a], Jackson Laboratory strain #002014) were lethally irradiated twice with 5 Gy using an X-RAD 320 X-Ray Biological Irradiator. $3 \times 10^7$ ER-Hoxb8 progenitors were washed twice with PBS to withdraw estrogen, resuspended in 200 μL of PBS, then delivered in a single retroorbital intravenous injection to irradiated mice. Tail vein incision blood samples of approximately 30 μL were obtained at timepoints after adoptive transfer. Samples were treated with red blood cell lysis buffer, washed in complete RPMI media, then stained for flow cytometry. ER-Hoxb8 progenitor-derived cells were identified as live CD45.1[+] CD45[+] CD3[-] B220- GFP[+] single cells.

## CRISPR pooled screens

The Brie library of murine sgRNAs on the lentiGuide-Puro backbone was purchased from Addgene and amplified by the UC Berkeley High Throughput Screening Facility. Protocols for amplification and experimental parameters were adapted from published protocols (*Joung et al., 2017*; *Sanjana et al., 2014*).sgRNAs were PCR amplified according to published protocols (*Doench et al., 2016*) and sequenced to verify sgRNA representation on the Illumina HiSeq 4000, 50 SR at the UC Berkeley Vincent J. Coates Genomics Sequencing Laboratory. ER-Hoxb8 neutrophil progenitors were transfected with lentivirus made with the sgRNA pooled library at an MOI of 0.11 to avoid multiple sgRNA transductions per cell, then selected with 5 μg/mL puromycin for 48 hr before freezing in liquid nitrogen. Pooled library cells were thawed into neutrophil progenitor medium containing β-estradiol and allowed to recover for 36–48 hr before differentiation and use in screens. A minimum of $4 \times 10^7$ progenitors were always maintained in either the differentiated or undifferentiated state in culture to retain at least 500 x coverage of sgRNAs during the screen. 5 ng/mL G-CSF was added to the differentiated cells at the start of β-estradiol withdrawal. ER-Hoxb8 neutrophil progenitors were passaged in non-treated T175 flasks and genomic DNA (gDNA) was harvested from $4 \times 10^7$ cells per condition using the Zymo gDNA Midiprep Kit. Purified gDNA was used for PCR amplification, library preparation with unique barcodes for each sample, and sequencing according to the Brie library sequencing protocol (*Doench et al., 2016*). Sequencing reactions were run at the UC Berkeley Vincent J. Coates

Genomics Sequencing Laboratory on an Illumina HiSeq 4000, 50 SR with 5% PhiX, single 8 bp index. Screen was performed in triplicate.

## CRISPR screen analysis

Sequencing results were deconvoluted and counts for each sgRNA were determined. The MAGeCK package (version 0.5.9.2) robust rank algorithm (RRA) was used to score all the genes in the Brie library (*Li et al., 2014*) by negative or positive enrichment in the experimental sample compared to the control sample. Enrichment analysis was done with default parameters with the exception of 'alphamean' for the log-fold-change method and Brie non-targeting sgRNAs as a control. MAGeCK-Flute (version 1.12.0) was used for gene set enrichment analysis using the top 200 most enriched genes and top 200 most depleted genes as determined by the MaGeCK package (the cutoff recommended in MAGeCKFlute documentation provided by bioconductor), with the following parameters: enrichment analysis method set to hypergeometric test and gene set type set to C5_CC (Gene Ontology Cellular Component terms as provided by the MSigDB gene collection). For validation experiments, the two sgRNA sequences with the greatest magnitude log fold change were cloned into pLGP and used to create single gene knockouts in ER-Hox8 progenitors.

## RNA-seq

$1 \times 10^6$ ER-Hoxb8 neutrophils transduced with a control non-targeting sgRNA were harvested at 0- and 2 days post estrogen withdrawal and G-CSF addition. Cells in suspension were harvested on day 0 and both floating and adherent cells were harvested on day 2 in 1 mL TRIzol (Invitrogen). Total RNA was extracted using chloroform, and the aqueous layer was further purified using RNeasy spin columns (Qiagen).

Library preparation and sequencing was performed by the QB3-Berkeley Genomics core labs at UC Berkeley. RNA quality was assessed on an Agilent 2100 Bioanalyzer and fragmented with an S220 Focused-Ultrasonicator (Covaris). Libraries were prepared using the KAPA RNA Hyper Prep kit (Roche). Truncated universal stub adapters were ligated to cDNA fragments, which were then extended via PCR using unique dual indexing primers into full-length Illumina adapters. Library quality was checked on an AATI (now Agilent) Fragment Analyzer. Library molarity was measured via quantitative PCR with the KAPA Library Quantification Kit (Roche) on a BioRad CFX Connect thermal cycler. Libraries were then pooled by molarity and sequenced on an Illumina NovaSeq 6000 S4 flowcell for 2x150 cycles, targeting at least 25 M reads per sample. Fastq files were generated and demultiplexed using Illumina bcl2fastq2 v2.20 and default settings, on a server running CentOS Linux 7. Data analysis was performed using FastQC for read quality assessment, Sythe, and Sickle for Illumina adapter and quality trimming, and TopHat2 for read alignment. Raw counts were derived from alignments using a STSeq-count Python script. Differential gene expression was performed with Seurat's FindMarkers function. Gene ontology analysis was performed with topGO (v 2.48.0).

## Acknowledgements

We thank David Sykes for invaluable advice on working with Hoxb8 immortalized neutrophils, Jacob Corn, Shaheen Kabir, and Allison Roberts for advice on CRISPR screening, Pingping He and Mary West at the High Throughput Screening Facility (UC Berkeley) for Brie library amplification and validation sample preparation, Justin Y Choi at the Functional Genomics Laboratory (UC Berkeley) for RNA sample quality check and CRISPR screen library sample preparation, QB3 Genomics and the Vincent J Coates Genomics Sequencing Laboratory (UC Berkeley, RRID:SCR_022170) for CRISPR library and screen sequencing, Reena Zalpuri at the Electron Microscope Laboratory (UC Berkeley) for advice and assistance in electron microscopy sample preparation and data collection, L Froenicke and the DNA Technologies and Expression Analysis Core (UC Davis) for RNAseq, and Matt Settles and Jie Li at the Genome Center and Bioinformatics Core Facility (UC Davis) for RNA-seq and data analysis. This work was supported by NIH S10 OD018174 Instrumentation Grant for the QB3 Genomics Sequencing Laboratory at UC Berkeley, the Kathleen L Miller Graduate Fellowship from the Henry Wheeler Center for Emerging and Neglected Diseases, NIH grant T32 GM 7232-40 and NSF Graduate Research Fellowship DGE-1752814 to RJ, and U19AI135990-01 to SS. We thank members of the Stanley and Cox labs for helpful discussions.

## Additional information

### Funding

| Funder | Grant reference number | Author |
|---|---|---|
| NIH Office of the Director | U19AI135990-01 | Sarah A Stanley |
| National Science Foundation Graduate Research Fellowship Program | DGE-1752814 | Robyn M Jong |
| NIH Office of the Director | T32 GM 7232-40 | Robyn M Jong |

The funders had no role in study design, data collection and interpretation, or the decision to submit the work for publication.

### Author contributions

Robyn M Jong, Conceptualization, Data curation, Formal analysis, Validation, Investigation, Visualization, Writing – original draft, Writing – review and editing; Krystal L Ching, Methodology, Writing – review and editing; Nicholas E Garelis, Formal analysis; Alex Zilinskas, Xammy Huu Wrynla, Sagar Rawal, Bianca C Hill, Bridget A Luckie, Lillian Shallow, Investigation; Jeffery S Cox, Formal analysis, Supervision; Gregory M Barton, Conceptualization, Supervision; Sarah A Stanley, Conceptualization, Resources, Data curation, Formal analysis, Supervision, Funding acquisition, Methodology, Writing – original draft, Project administration, Writing – review and editing

### Author ORCIDs

Krystal L Ching ⓘ https://orcid.org/0000-0002-1181-0119
Xammy Huu Wrynla ⓘ https://orcid.org/0000-0002-7532-8356
Bridget A Luckie ⓘ https://orcid.org/0000-0003-3754-2375
Jeffery S Cox ⓘ https://orcid.org/0000-0002-5061-6618
Gregory M Barton ⓘ https://orcid.org/0000-0002-3793-0100
Sarah A Stanley ⓘ https://orcid.org/0000-0002-4182-9048

### Ethics

All procedures involving the use of mice were approved by the University of California, Berkeley Institutional Animal Care and Use Committee (protocol 2015-09-7979). All protocols conform to federal regulations, the National Research Council Guide for the Care and Use of Laboratory Animals, and the Public Health Service Policy on Humane Care and Use of Laboratory Animals.

### Decision letter and Author response

Decision letter https://doi.org/10.7554/eLife.82289.sa1
Author response https://doi.org/10.7554/eLife.82289.sa2

---

## Additional files

### Supplementary files

Supplementary file 1. Gene expression profiles comparing day 0 and day 2 post estrogen withdrawal for Cas9+ER-Hoxb8 cells.

Supplementary file 2. MAGeCK RRA scores for genes positively and negatively selected during differentiation of Cas9+ER-Hoxb8 progenitors.

Transparent reporting form

### Data availability

RNAseq data is uploaded into GEO: GSE211699 Processed RNAseq data has been included as a supplemental attachment. CRISPR screening data is included as a supplemental attachment. All data generated or analyses during this study are included in the manuscript and supporting file.

The following dataset was generated:

| Author(s) | Year | Dataset title | Dataset URL | Database and Identifier |
|---|---|---|---|---|
| Stanley S, Jong R, Rawal S | 2022 | Transcriptional profile of Cas9+HoxB8 neutrophil progenitors at day 0 and day 2 after withdrawal of estrogen | https://www.ncbi.nlm.nih.gov/geo/query/acc.cgi?acc=GSE211699 | NCBI Gene Expression Omnibus, GSE211699 |

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
