## [Editor Report]

Jong et al. provide and convincingly validate a resource for performing CRISPR screenings to study neutrophil differentiation and function by generating Hoxb8 cells that constitutively express Cas9. This valuable library-screening approach has the potential to improve on the established lentiviral CRISPR-Cas9 editing of Hoxb8 cells.

---

## [Decision Letter]

**Decision letter after peer review:**

Thank you for submitting your article "Cas9^+^ conditionally immortalized neutrophil progenitors as a tool for genome wide CRISPR screening for neutrophil differentiation and function" for consideration by *eLife*. Your article has been reviewed by 3 peer reviewers, and the evaluation has been overseen by a Reviewing Editor and Carla Rothlin as the Senior Editor. The following individual involved in the review of your submission has agreed to reveal their identity: Alejo Rodriguez-Fraticelli (Reviewer #2).

*Essential Revisions:*

1) It is unclear whether the technical achievement is impactful enough: in vivo genome-wide screening, improvement in efficiency compared to published protocols with Cas9 delivery in plasmids, and validation of the role of identified hits in neutrophil maturation itself should be included to reinforce the strength of this tool for the community.

2) Key references, a more explicit explanation of the importance of constant Cas9 expression in Hoxb8 cells, a more detailed representation of results, and thoughtful discussion regarding the novelty and impact of the tool beyond the validation of the cell line should be improved/included.

*Reviewer #1 (Recommendations for the authors):*

Identified candidates associated with neutrophil differentiation, as those indicated in Figure 4A, were not validated in vivo using neutrophil-specific K.O. models or further characterized in vitro (e.g. transcriptional or epigenetic changes during maturation when compared to non-targeting sgRNA controls). Could authors provide further evidence on how some of these newly identified candidates could impact the biology of neutrophils?

It has been already proved that Hoxb8 neutrophil progenitors are able to give rise to cells that resemble a neutrophil. The authors test this in their study and provide a good characterization of the differentiation process of Cas9+ER-Hoxb8 cells, Specially the TEM images are clearly indicative of a changing nuclear morphology toward a mature phenotype. In addition, the authors use a transcriptional analysis of DEGs at D0 prior to estrogen withdrawal and on day 2 post estrogen withdrawal in the presence of G-CSF to generate a gene signature to explore its transcriptional similarity to a neutrophil. Why authors used D2 instead of D4 post estrogen withdrawal? This should have given better results. In addition, this analysis could be further strengthened using available public databases and integration methods to demonstrate that point.

Although this technique has great potential in the field the study does not go beyond the state of the art. I suggest that coupling this technological platform with single-cell transcriptional analysis could generate a more robust platform to explore the effect of regulatory gene networks in neutrophils by large-scale genetic screens.

Authors should discuss the limitations of working with this in vitro system as Hoxb8 cells may not recapitulate neutrophil phenotype in vivo. Also, it would be interesting to develop further the in vivo transfer protocol (e.g. use other mutants by FACs analysis or perform cell sorting of the transfer cells for cytospins or transcriptional analysis).

*Reviewer #2 (Recommendations for the authors):*

While I find this resource will be very useful for the community of neutrophil researchers, I also think the methodological achievement is incremental. Hoxb8-immortalization of progenitors was described 20 years ago, and it has been used on and on for multiple studies, far more in-depth than this one. It is also well established that the Rosa26-Cas9 mouse model works efficiently in hematopoietic and hematopoietic-derived immune cells (LaFleur et al. Nat Comm 2019). Thus, it is unclear to me whether the technical achievement is impactful enough.

Exploration of the potential of this technique for in vivo genome-wide screening in neutrophils seems like the most relevant thing to take a look at. It would be amazing if the library representation in vivo is maintained enough to allow for genome-wide screening. This would be a total breakthrough for the field (as current implementations fully depend on HSC/BM engraftment, which leads to a reduced representation of guides per mouse).

Alternatively, I would suggest at least some basic mechanistic interrogation of key hits. A simple imaging/morphological study and a gene-expression (RNAseq) study, which are two techniques used by the authors during the validation steps (Figure 1-3), would go a long way. This would show which stage of differentiation is being blocked and help authors at least suggest a few mechanisms to follow up for the newly identified regulator complexes.

In sum, while the manuscript, as put together in the current version, falls a bit short, I do believe the resource is super useful for the community and it could have a great impact in the field.

*Reviewer #3 (Recommendations for the authors):*

1. Key references are lacking in this manuscript, especially the papers by Wang et al. [Wand et al., JCI insight 2020] and Khoyratty et al. [Khoyratty et al., Nat Imm 2021], where they thoroughly characterised the stages of HoxB8 differentiation in culture and genetically modify HoxB8 progenitor cells by lentiviral CRISPR-Cas9 and study the function of several TFs in neutrophil maturation and effector functions. Phagocytosis and bacterial killing by Hoxb8 neutrophils have also been described and need to be referenced.

2. The manuscript would benefit from a more explicit explanation of the importance of constant Cas9 expression in Hoxb8 cells. It needs to state clearly how this technique outperforms the published approach of introducing Cas9 via plasmid together with gRNA.

3. As HoxB8 has been characterised in detail previously, the characterisation of Cas9+ Hoxb8 cells falls into the technical aspect of the work. So, the rationale for including these in the main figures could be either to compare them to the original wt HoxB8 cells to prove that Cas9+- derived HoxB8 neutrophils are largely similar; or to compare different HoxB8 KO lines for the same target – i.e. clones of CEBPE generated here and previously with Cas9 introduced by a plasmid packaged into lentivirus [Khoyratty et al., Nat Imm 2021].

4. In CEBPE KO transfer experiments two improvements can be done to strengthen the data and conclusions. The first is to account for possible cell death of CEBPE KO cells by assessing their numbers in comparison to wt in addition to 5 of mature cells. The second is to account for possible differences in the recipient environment, by introducing wt and CEBPE KO cells labelled with different fluorophores into the same animal.

5. The authors should expand on the comprehensive characterisation of the hits from genome wide-screening. The library screen is the most interesting result of the study. There is a discrepancy in the subtitle (screen for survival) and the text (screen for regulators of maturation). Ideally, phenotypic, functional, and morphological characterization of generated KO lines (at least WASH, CEBPe) should have been included to further validate the role of these hits in neutrophil survival vs maturation.

[Editors' note: further revisions were suggested prior to acceptance, as described below.]

Thank you for submitting your article "Cas9^+^ conditionally immortalized neutrophil progenitors as a tool for genome wide CRISPR screening for neutrophil differentiation and function" for consideration by *eLife*. Your article has been reviewed by 1 peer reviewers at Review Commons, and the evaluation at *eLife* has been overseen by a Reviewing Editor and Carla Rothlin as the Senior Editor.

The manuscript has been improved but there are some remaining issues that need to be addressed, as outlined below:

We recommend including a discussion on the limitations of the study given the reviewers' remaining concern.

*Reviewer 1:*

I broadly agree with the assessments provided by the other reviewers. While all reviewers acknowledge that the system described may be a useful resource for the community, there is a consistent concern regarding the incremental nature of the advance and the limited functional or mechanistic validation of the CRISPR screening results. Overall, the reviews converge on the view that the technical platform is sound, but that the evidence presented does not yet support strong biological or conceptual claims beyond resource development.

*Reviewer 2:*

My comments have been addressed by authors, and they have amended the text and provided additional nuance to the discussion as recommended by multiple reviewers. I have no further concerns to remark.

---

## [Author Response]

Essential Revisions:1) It is unclear whether the technical achievement is impactful enough: in vivo genome-wide screening, improvement in efficiency compared to published protocols with Cas9 delivery in plasmids, and validation of the role of identified hits in neutrophil maturation itself should be included to reinforce the strength of this tool for the community.2) Key references, a more explicit explanation of the importance of constant Cas9 expression in Hoxb8 cells, a more detailed representation of results, and thoughtful discussion regarding the novelty and impact of the tool beyond the validation of the cell line should be improved/included.Reviewer #1 (Recommendations for the authors):Identified candidates associated with neutrophil differentiation, as those indicated in Figure 4A, were not validated in vivo using neutrophil-specific K.O. models or further characterized in vitro (e.g. transcriptional or epigenetic changes during maturation when compared to non-targeting sgRNA controls). Could authors provide further evidence on how some of these newly identified candidates could impact the biology of neutrophils?

We think this is an important point. Investigating these newly identified candidates is the subject of future studies by our own group and we hope, by other labs. Our primary goal with this publication is to present this as a resource to the community.

It has been already proved that Hoxb8 neutrophil progenitors are able to give rise to cells that resemble a neutrophil. The authors test this in their study and provide a good characterization of the differentiation process of Cas9+ER-Hoxb8 cells, Specially the TEM images are clearly indicative of a changing nuclear morphology toward a mature phenotype. In addition, the authors use a transcriptional analysis of DEGs at D0 prior to estrogen withdrawal and on day 2 post estrogen withdrawal in the presence of G-CSF to generate a gene signature to explore its transcriptional similarity to a neutrophil. Why authors used D2 instead of D4 post estrogen withdrawal? This should have given better results. In addition, this analysis could be further strengthened using available public databases and integration methods to demonstrate that point.

We chose to profile day 2 so that we could capture differentiation for possible comparison to mutants that have differences in their differentiation profiles.

Although this technique has great potential in the field the study does not go beyond the state of the art. I suggest that coupling this technological platform with single-cell transcriptional analysis could generate a more robust platform to explore the effect of regulatory gene networks in neutrophils by large-scale genetic screens.

We appreciate this comment and hope to perform scRNAseq using this system in the future.

Authors should discuss the limitations of working with this in vitro system as Hoxb8 cells may not recapitulate neutrophil phenotype in vivo. Also, it would be interesting to develop further the in vivo transfer protocol (e.g. use other mutants by FACs analysis or perform cell sorting of the transfer cells for cytospins or transcriptional analysis).

We have added this to the discussion as requested.

Reviewer #2 (Recommendations for the authors):While I find this resource will be very useful for the community of neutrophil researchers, I also think the methodological achievement is incremental. Hoxb8-immortalization of progenitors was described 20 years ago, and it has been used on and on for multiple studies, far more in-depth than this one. It is also well established that the Rosa26-Cas9 mouse model works efficiently in hematopoietic and hematopoietic-derived immune cells (LaFleur et al. Nat Comm 2019). Thus, it is unclear to me whether the technical achievement is impactful enough.Exploration of the potential of this technique for in vivo genome-wide screening in neutrophils seems like the most relevant thing to take a look at. It would be amazing if the library representation in vivo is maintained enough to allow for genome-wide screening. This would be a total breakthrough for the field (as current implementations fully depend on HSC/BM engraftment, which leads to a reduced representation of guides per mouse).

We agree that in vivo screening will be a breakthrough for the field, and hope that this study inspires other investigators to test and utilize this resource for such studies.

Alternatively, I would suggest at least some basic mechanistic interrogation of key hits. A simple imaging/morphological study and a gene-expression (RNAseq) study, which are two techniques used by the authors during the validation steps (Figure 1-3), would go a long way. This would show which stage of differentiation is being blocked and help authors at least suggest a few mechanisms to follow up for the newly identified regulator complexes.

Our belief is that these data will be a resource for people in the wider scientific community with specific interest in neutrophils to inspire such mechanistic studies in the future.

In sum, while the manuscript, as put together in the current version, falls a bit short, I do believe the resource is super useful for the community and it could have a great impact in the field.

We thank the reviewer for this comment.

Reviewer #3 (Recommendations for the authors):1. Key references are lacking in this manuscript, especially the papers by Wang et al. [Wand et al., JCI insight 2020] and Khoyratty et al. [Khoyratty et al., Nat Imm 2021], where they thoroughly characterised the stages of HoxB8 differentiation in culture and genetically modify HoxB8 progenitor cells by lentiviral CRISPR-Cas9 and study the function of several TFs in neutrophil maturation and effector functions. Phagocytosis and bacterial killing by Hoxb8 neutrophils have also been described and need to be referenced.

We have added these references to the manuscript.

2. The manuscript would benefit from a more explicit explanation of the importance of constant Cas9 expression in Hoxb8 cells. It needs to state clearly how this technique outperforms the published approach of introducing Cas9 via plasmid together with gRNA.

We believe that techniques that employ Cas9 delivery with sgRNA are complementary to one another; one technique need not outperform the other, and the choice of technique to use will be individual to the investigator and biological problem of interest.

3. As HoxB8 has been characterised in detail previously, the characterisation of Cas9+ Hoxb8 cells falls into the technical aspect of the work. So, the rationale for including these in the main figures could be either to compare them to the original wt HoxB8 cells to prove that Cas9+- derived HoxB8 neutrophils are largely similar; or to compare different HoxB8 KO lines for the same target – i.e. clones of CEBPE generated here and previously with Cas9 introduced by a plasmid packaged into lentivirus [Khoyratty et al., Nat Imm 2021].

Our rationale for describing these cells in detail here is to demonstrate that the Cas9+ cells do in fact behave the same as the original cells reported.

4. In CEBPE KO transfer experiments two improvements can be done to strengthen the data and conclusions. The first is to account for possible cell death of CEBPE KO cells by assessing their numbers in comparison to wt in addition to 5 of mature cells. The second is to account for possible differences in the recipient environment, by introducing wt and CEBPE KO cells labelled with different fluorophores into the same animal.

We did evaluate numbers of cells and saw the expected differences; however this data was noisier/variable which is why we chose to present percentages. We thank the reviewer for the suggestion of utilizing different fluorophores which can be adopted in future experiments.

5. The authors should expand on the comprehensive characterisation of the hits from genome wide-screening. The library screen is the most interesting result of the study. There is a discrepancy in the subtitle (screen for survival) and the text (screen for regulators of maturation). Ideally, phenotypic, functional, and morphological characterization of generated KO lines (at least WASH, CEBPe) should have been included to further validate the role of these hits in neutrophil survival vs maturation.

We have corrected the discrepancies and added further detail to the text.

[Editors’ note: what follows is the authors’ response to the second round of review.]

The manuscript has been improved but there are some remaining issues that need to be addressed, as outlined below:We recommend including a discussion on the limitations of the study given the reviewers' remaining concern.

We have now added a paragraph in the discussion that begins “It is important to note, however, that there are some limitations of this study”. In this paragraph we include the following text:

“Second, this study primarily presents a technical resource that demonstrates the feasibility of using HoxB8 Cas9+ neutrophils to identify novel regulators of neutrophil differentiation and survival. Further investigation is necessary to clearly establish a biological/mechanistic role for genes we identified in our screens for neutrophil differentiation, both in vivo and in vitro. Future work on the hits described could lead to new tools for manipulating neutrophil numbers in vivo as and for understanding the pathways that regulate their survival.”

Reviewer 1:I broadly agree with the assessments provided by the other reviewers. While all reviewers acknowledge that the system described may be a useful resource for the community, there is a consistent concern regarding the incremental nature of the advance and the limited functional or mechanistic validation of the CRISPR screening results. Overall, the reviews converge on the view that the technical platform is sound, but that the evidence presented does not yet support strong biological or conceptual claims beyond resource development.

Please see the above comment.

Reviewer 2:My comments have been addressed by authors, and they have amended the text and provided additional nuance to the discussion as recommended by multiple reviewers. I have no further concerns to remark.

We thank the reviewer!